# Tannic-Acid-Cross-Linked and TiO_2_-Nanoparticle-Reinforced Chitosan-Based Nanocomposite Film

**DOI:** 10.3390/polym13020228

**Published:** 2021-01-11

**Authors:** Swarup Roy, Lindong Zhai, Hyun Chan Kim, Duc Hoa Pham, Hussein Alrobei, Jaehwan Kim

**Affiliations:** 1Creative Research Center for Nanocellulose Future Composites, Department of Mechanical Engineering, Inha University, Incheon 22212, Korea; swaruproy2013@gmail.com (S.R.); duicaofei@naver.com (L.Z.); Kim_HyunChan@naver.com (H.C.K.); phamduchoa.tdt@gmail.com (D.H.P.); 2Department of Mechanical Engineering, Prince Sattam bin Abdul Aziz University, AlKharj 11942, Saudi Arabia; h.alrobei@psau.edu.sa

**Keywords:** chitosan, tannic acid, titanium dioxide, nanocomposite film, mechanical properties, antioxidant activity

## Abstract

A chitosan-based nanocomposite film with tannic acid (TA) as a cross-linker and titanium dioxide nanoparticles (TiO_2_) as a reinforcing agent was developed with a solution casting technique. TA and TiO_2_ are biocompatible with chitosan, and this paper studied the synergistic effect of the cross-linker and the reinforcing agent. The addition of TA enhanced the ultraviolet blocking and mechanical properties of the chitosan-based nanocomposite film. The reinforcement of TiO_2_ in chitosan/TA further improved the nanocomposite film’s mechanical properties compared to the neat chitosan or chitosan/TA film. The thermal stability of the chitosan-based nanocomposite film was slightly enhanced, whereas the swelling ratio decreased. Interestingly, its water vapor barrier property was also significantly increased. The developed chitosan-based nanocomposite film showed potent antioxidant activity, and it is promising for active food packaging.

## 1. Introduction

Nowadays, the use of synthetic non-biodegradable plastics in packaging areas has become an ample hazard for our environment. In the year 2015, it was reported that about 6.3 billion tons of plastic waste were formed worldwide [1], and according to the Environmental Protection Agency (EPA), about 40% of municipal waste is generated from plastic packaging [2]. In this context, to overcome the recent plastic-based packaging problem, the effort to develop biodegradable plastics based on biopolymers has gained proper attention in order to replace synthetic polymer-based plastics [3,4,5,6,7,8]. A recent report showed that bioplastics in the European market have now replaced ~10% of the present plastic market. The European Parliament set a target of using 100% reusable plastic by 2030 [9,10]. Biopolymers have many advantages, including biodegradability, renewability, biocompatibility, and eco-friendliness [6,11,12,13,14,15]. Biopolymers have been extensively used to make films. Chitosan is one of the most promising biopolymers because of its excellent antimicrobial activity [16,17,18,19]. Chitosan is a linear polysaccharide formed after chitin’s de-acetylation. Chitosan has fair uses in food packaging and product storage, even though weak mechanical and barrier properties restrict its bulk use as an alternative to synthetic plastics [20,21,22,23,24]. Chitosan-based film’s limitations can be improved using chemical cross-linkers and physical reinforcement with nanofillers [18,25]. A previous study was carried out based on physical cross-linking of chitosan to improve its physical properties [26,27,28]. Additionally, varieties of nanofillers and bioactive functional compounds, such as chitin nanowhiskers, cellulose nanofibers, metal nanoparticles (inorganic and organic), essential oils, etc., have already been reinforced to improve the physical and functional characteristics of chitosan-based film [15,29,30,31,32,33,34,35,36,37].

In this context, many chemical cross-linkers, such as glutaraldehyde, formaldehyde, glyoxal, tannic acid, vanillin, citric acid, genipin, and tripolyphosphate, have been used to improve chitosan films’ physical and functional properties [30,38,39]. Tannic acid (TA) has received attention as it is a natural polymer, non-cytotoxic, and widely used in food applications [26]. TA is a water-soluble, polyphenolic, and amphiphilic polymer with excellent cross-linking and antioxidant proficiency [39,40]. On the other hand, different metal nanoparticles and metal oxide nanoparticles have also been used as chemical reinforcing agents so far [41,42]. Among the metal oxide nanoparticles, titanium dioxide (TiO_2_) has recently been utilized to prepare different composites. It is inert, non-toxic, eco-friendly, and inexpensive, with excellent biocompatibility [10,43]. The incorporation of TiO_2_ is known to improve various physical properties, such as the mechanical, barrier, UV-light barrier, thermal, optical, and antimicrobial properties of composite films [10,44,45]. Previously, TA was used to prepare chitosan-based film [26,39,46]. The effect of TiO_2_ on chitosan-based film was also studied [31,47]. To the best of our knowledge, the synergistic effect of TA and TiO_2_ in chitosan-based nanocomposite films has not been studied so far. Previous works mostly focused only on cross-linking chitosan with TA or chitosan and various TiO_2_ nanoparticles.

Moreover, the cross-linker’s concentration and the TiO_2_ nanoparticles’ size also significantly impact composite films’ properties. In the present work, spherical-shape TiO_2_ nanoparticles and the optimum TA content were used as reinforcing agents and a cross-linker, respectively. More specifically, this is a more in-depth study of TA-cross-linked and TiO_2_-reinforced chitosan nanocomposite film. However, the various properties were not thoroughly studied in previous studies, such as the water vapor barrier properties, hydrodynamic properties, and antioxidant activity. Combining a nano-filler and cross-linker in chitosan-based composite film is expected to improve the film’s physical and functional properties. This results obtained in this study are expected to produce new insight into chitosan-based functional composite films for active food packaging applications.

For the fabrication of biobased, eco-friendly, and biodegradable active packaging film, many techniques, such as solution casting, compression molding, extrusion (casting and blowing), etc., are commonly used [48,49]. Solution casting is the most convenient and straightforward method for producing the film on a lab scale. The solution mixing and casting methods have a low cost, are easy to handle, and are efficient for biopolymer-based packaging films [50]. Even though the solution casting method is not useful for industrial-scale production, the ideal composition prepared in this way can be used to further develop environmentally benign packaging films via extrusion methods for mass production.

The present investigation’s primary objective is to make a chitosan-based nanocomposite film with improved mechanical and functional properties using a TA cross-linker and a physical reinforcement by TiO_2_. The developed nanocomposite film was prepared with a simple solution mixing and casting technique and was characterized using various analytical methods. The film properties, such as the mechanical properties, thermal stability, hydrodynamic properties, water vapor barrier properties, and antioxidant activity, were also assessed.

## 2. Materials and Methods

### 2.1. Materials

Chitosan (Chs) (viscosity 200–800 cP at 1% acetic acid, MW: 190,000–310,000 based on viscosity, 75–85% deacetylated), titanium oxide nanopowder, dopamine hydrochloride, sodium hydroxide, 2,2-diphenyl-1-picrylhydrazyl (DPPH), 2,2′-azino-bis(3-ethylbenzothiazoline-6-sulfonic acid) (ABTS), and potassium persulfate were purchased from Sigma-Aldrich, St. Louis, MO, USA. All other chemicals used were of analytical reagent grade.

### 2.2. Preparation of Chitosan/TA/TiO_2_ Nanocomposite Films

The chitosan-based nanocomposite films were prepared by following the solution casting method [15]. Initially, the chitosan solution was designed using 0.5% acetic acid [15,51], and 5 wt% (based on chitosan) of TA was mixed, followed by mixing 0.5 and 1.0 wt% (based on chitosan) of TiO_2_ with the prepared chitosan solution at room temperature. The mixture was pulverized using a high-shear homogenizer (T50, IKA Labotechnik, Germany) for 10 min at 5000 rpm, followed by sonication for another 1 h. The completely soluble film-forming solution was cast on a polycarbonate plate using a doctor blade and dried for 48 h in a temperature- and humidity-controlled cleanroom (25 °C and 50% RH). The dried film was peeled off from the plate and left in the cleanroom for 48 h. For comparison, a neat chitosan film without any additives was also prepared by following the same procedure. All film samples were made in triplicate, and the developed films were designated as Chs/TA, Chs/TA/Ti^0.5^, and Chs/TA/Ti^1.0^, respectively, as per the content of additives. The preparation of the nanocomposite film is briefly explained in the following schematics (Scheme 1).

### 2.3. Characterization

#### 2.3.1. Morphology

The surface and cross-sectional morphologies of the chitosan-based nanocomposite films and the morphology of TiO_2_ were checked with field-emission scanning electron microscopy (FESEM, SU8010, Hitachi, Tokyo, Japan) at accelerating voltages of 5 and 15 kV, respectively. All the film specimens and the TiO_2_ powder were sputter-coated with platinum for 90 s before the measurement.

#### 2.3.2. FTIR and Optical Properties

The Fourier-transform infrared (FTIR) spectra of the chitosan-based nanocomposite films were noted in an FTIR spectrometer (Billerica, Bruker Optics) in attenuated total reflection (ATR) mode in the range of 4000–650 cm^−1^ at 16 scan rates with the resolution of 4 cm^−1^. The nanocomposite films’ optical properties were recorded using a UV-vis spectrophotometer (UV-2501PC, Shimadzu, Kyoto, Japan) in 200–800 nm. The UV-barrier property and transparency of the same films were also evaluated by determining the transmittance at 280 nm (T_280_) and 660 nm (T_660_), respectively [12].

#### 2.3.3. Moisture Content, Water Solubility, and Swelling Ratio

The moisture content (MC), water solubility (WS), and swelling ratio (SR) of the chitosan-based nanocomposite films were determined by following the standard method [15]. The MC of the specimens (2.5 × 2.5 cm) was measured as the film’s weight transformation after dehydrating at 105 °C for 24 h. The MC was calculated using the following equation:(1)$$\mathrm{MC}\;\left(\%\right)=\;\frac{W_{1}-W_{2}}{W_{1}}\times 100$$
where *W*_1_ and *W*_2_ refer to the initial and dried weight of the film specimens, respectively.

For WS, at first, the tested film specimens (2.5 × 2.5 cm) were dried at 60 °C overnight and weighed (*W_1_*), which was followed by dipping in 30 mL of deionized (DI) water for 24 h with occasional mild shaking at 25 °C; then, the specimens were taken out from the water, dried in an oven at 105 °C for 24 h, and then weighed (*W*_2_). The WS was calculated using the following equation:(2)$$\mathrm{WS}\;\left(\%\right)=\;\frac{W_{1}-W_{2}}{W_{1}}\times 100$$

To determine the SR, a pre-weighed (*W*_1_) film specimen (2.5 × 2.5 cm) was immersed in 30 mL DI water for 1 h; then, it was taken out of the water and weighed (*W*_2_) after removal of surface water using a blotting paper. The SR was calculated using the following equation:(3)$$\mathrm{SR}\;\left(\%\right)=\;\frac{W_{2}-W_{1}}{W_{1}}\times 100$$

#### 2.3.4. Water Vapor Permeability

The water vapor permeability (WVP) of the chitosan-based nanocomposite films was determined using a WVP cup by following the ASTM E96-95 standard method. At first, the WVP cup was filled with a prescribed amount of water, covered by the films, sealed, and kept in the controlled environmental chamber at 25 °C and 50% RH. After equilibration, the WVP cup’s weight was measured at every 1 h interval, and weight loss was calculated. The water vapor transmission rate can be measured from the ratio of weight loss and the film area. The WVP (g.m/m^2^.Pa.s) was determined as follows:(4)$$\mathrm{WVP}=\frac{\Delta W\times L}{t\times A\times \Delta p}$$
where Δ*W* is the weight alteration of the WVP cup (g), *L* is the thickness of the film (m), ∆*p* is the partial water vapor pressure difference across the two sides of the film, *A* is the permeation area of the film (m^2^), and *t* is the time (s). The same was calculated using the established procedure [52].

#### 2.3.5. Mechanical Properties

The film specimens’ thickness was measured using a digital micrometer (Digimatic, Mitutoyo, Kawasaki, Japan) with 1 mm accuracy. For each specimen, two random positions were taken, and the average values were used. Mechanical properties, namely the tensile strength (TS), Young’s modulus (YM), and elongation at break (EB), of the prepared nanocomposite films were measured according to the standard (ASTM D-882-97) with a household tensile testing system [53]. The tensile test was conducted in a controlled environment at 25 °C and ~30% RH. The sizes of the film specimens were 5 × 1 cm, and the gauge length and applied pulling rate were 20 mm and 0.005 mm/s, respectively; four specimens were tested for each case, and the values were averaged.

#### 2.3.6. Differential Scanning Calorimetry and Thermogravimetric Analysis

Thermogravimetric analysis (TGA) and differential scanning calorimetry (DSC) were performed to understand the thermal properties. The TGA measurement was carried out using a TGA (STA 409 PC, Netzsch, Selb, Germany), and ~10 mg film specimens were tested at a heating rate of 10 °C/min in a temperature range of 30 to 600 °C under a nitrogen flow of 20 cm^3^/min. The maximum disintegration temperature was determined from differential thermogravimetry (DTG) curves [54]. The DSC was observed using a TA instrument (DSC200 F3, Netzsch) at a heating rate of 10 °C/min in a temperature range of 20 to 350 °C under a nitrogen gas atmosphere.

#### 2.3.7. Antioxidant Activity

Antioxidant activities of the chitosan-based nanocomposite films were measured by assessing the free radical scavenging activity. The 2,2-diphenyl-1-picrylhydrazyl radical (DPPH^•^) and 2,2′-azino-bis(3-ethylbenzothiazoline-6-sulfonic acid) (ABTS^•+^) radical scavenging methods [55,56] were used for the antioxidant test. For the DPPH analysis, a prescribed amount of methanolic solution of DPPH was freshly made, and ~50 mg of the tested film sample was added in a 10 mL DPPH solution and incubated at room temperature for 30 min; then, the absorbance was measured at 517 nm. A control was also tested without adding the film sample in an assay solution. In the ABTS assay, a prescribed amount of potassium sulfate was added to the ABTS solution, followed by overnight incubation in the dark to make the ABTS assay solution. A total of ~50 mg of tested film samples was added to 10 mL of ABTS assay solution and incubated at room temperature for 30 min; then, the absorbance was measured at 734 nm. A control was also tested without adding the film sample in an assay solution. The antioxidative activity was calculated as follows:(5)$$\mathrm{Free}\;\mathrm{radical}\;\mathrm{scavenging}\;\mathrm{activity}\;\left(\%\right)=\;\frac{A_{c}-A_{t}}{A_{c}}\times 100$$
where *A_c_* and *A_t_* were the absorbances of DPPH/ABTS of the control and test films, respectively.

#### 2.3.8. Antioxidant Activity

For statistical analysis of the obtained results, one-way analysis of variance (ANOVA) was performed, and the significance of each mean property value was determined (*p* < 0.05) by Duncan’s multiple range test using the SPSS statistical analysis computer program for Windows (SPSS Inc., Chicago, IL, USA).

## 3. Results and Discussion

### 3.1. Properties of TiO_2_ Nanoparticles

Figure 1a shows the UV-vis absorption of aqueous TiO_2_ solution, and the results obtained from the absorption spectrum showed a distinct absorption profile for TiO_2_ with a maximum absorption of ~350 nm. The detected UV-vis results are physical characteristics of TiO_2_, and the obtained results corroborate the findings of a previously published report [57]. The morphology of the TiO_2_ taken by FESEM is shown in Figure 1b. The TiO_2_ was roughly spherical and in the size range of 15–45 nm, with an average diameter of 31.3 ± 5.8 nm, as determined with the ImageJ software.

### 3.2. Properties of Chitosan/TA/TiO_2_ Nanocomposite Films

#### 3.2.1. Appearance and Optical Properties

The macroscopic appearance of chitosan-based nanocomposite film is displayed in Figure 2a. The neat chitosan film was freestanding, flexible, highly transparent, and colorless. In contrast, the nanocomposite film containing TA exhibited a light brownish color, and the blended TiO_2_ films were dark brown depending on the content of TiO_2_. The optical properties of the chitosan, Chs/TA, and Chs/TA/TiO_2_ nanocomposite films are shown in Figure 2b. The UV-vis spectra exhibit that the neat chitosan film is highly transparent, and in the case of nanocomposite films, the profile nature is different. For better understanding, the UV-light barrier and transparency of the chitosan-based nanocomposite films were scrutinized by measuring the absorbance at 280 and 660 nm. The results are displayed in Table 1. The UV-light transmittance and transparency of the neat chitosan film were 52.5% and 86.3%, respectively, similarly to the previously published report [15]. The UV-light barrier property was almost completely blocked after adding the TA in chitosan, which might be due to TA’s strong UV-light absorption [58]. After adding TiO_2,_ the UV-light barrier remained practically unaltered. The observed results suggest that by adding TA in the chitosan-based nanocomposite films, the UV light can be blocked entirely, benefitting their application in active packaging. The neat chitosan film’s transparency was slightly reduced after adding TA, whereas the incorporation of TiO_2_ significantly decreased the transparency depending on the nanoparticle content. These results suggest that TA’s addition slightly decreases transparency, but addition of TiO_2_ reduces the transparency significantly, although well enough for packaging film applications.

#### 3.2.2. Fourier−Transform Infrared Spectroscopy

The FTIR spectra of the neat chitosan and its nanocomposite films are presented in Figure 3. The peak appeared in the range of 3600–3000 cm^−1^, mainly due to the O-H and N-H stretching vibrations [15]. The peak observed at 2877 cm^−1^ was attributed to the C-H stretching vibrations of the alkane groups of chitosan. Peaks found at 1645 cm^−1^ and 1548 cm^−1^ referred to the C=O stretching of the acetyl group (amide-I) of chitosan and -NH bending and stretching (amide-II), respectively [59]. Peaks noticed at 1405 cm^−1^ and ~1151 cm^−1^ were ascribed to the O-H bending vibration due to chitosan’s saccharide structure [15]. In chitosan-based nanocomposite films, similar peak profiles with a slight alteration in intensity and positions were observed. In conclusion, there was no characteristic alteration in the chitosan-based films’ functional groups after adding the cross-linker and reinforcing agent. The minor modifications in intensity and position in the peaks might be because of the physical interaction (H-bonding, van der Waals forces) between chitosan and TA/TiO_2_ [5].

#### 3.2.3. Morphology

The microstructure (surface and cross-section morphology) of the prepared chitosan-based nanocomposite films found by scanning electron microscopy (SEM) is shown in Figure 4a–h. The surface images of the neat chitosan and its nanocomposite films indicate that all of the films are freestanding, smooth, and without any cracks or voids (Figure 4a–d). The addition of TA/TiO_2_ did not meaningfully alter the dense structure of the surface morphology. The incorporation of TA/TiO_2_ did not alter the morphology, and they were uniformly mixed in the matrix polymer, which indicates their good miscibility in the liquid phase. The cross-section morphologies of the chitosan-based films exhibited that the neat chitosan film had a dense layer structure with some cracks.

In contrast, the blended TA/TiO_2_ films showed similar morphologies, but a more porous structure due to the tannic acid (Figure 4e–h). The addition of TiO_2_ at 1 wt% showed some ruptures and voids in the chitosan matrix’s layered structure. The obtained results indicate that the TA/TiO_2_ is compatible with the chitosan matrix, and that the fillers are uniformly distributed in the chitosan matrix. These results show the excellent adhesion, intermolecular binding, and affinity between chitosan and TA/TiO_2_, which might play an essential role in improving the physical properties.

#### 3.2.4. Thermal Properties

The chitosan nanocomposite film’s thermal stability was assessed by measuring TGA and DTG thermograms and DSC analysis. The TGA and DTG results are displayed in Figure 5a,b. It was observed that both chitosan and its nanocomposite films show a twofold thermal degradation arrangement. The maximum initial weight loss was demonstrated at ~75–80 °C for all the tested films due to the vaporization of moisture [15]. The onset/endset temperature for the primary degradation was observed at 144/390 °C (Table 1). The maximum thermal degradation was observed around 270–280 °C due to the thermal decomposition of the chitosan polymer matrix [15]. The maximum degradation temperature for the neat chitosan was 270 °C, whereas, after the addition of TA, it slightly increased to 273 °C. The addition of TiO_2_ slightly increased the thermal decomposition temperature of the chitosan-based films (Table 1). It was recently reported that the addition of TiO_2_ in biopolymer-based composite film improved thermal stability [43]. The char contents observed at 600 °C for neat chitosan, Chs/TA, Chs/TA/Ti^0.5^, and Chs/TA/Ti^1.0^ films were 41.2, 43.4, 41.2, and 42.8%, respectively, which is at the higher side, and most probably comes from the non-ignitable mineral present in the biopolymer used. The DSC spectra of the neat chitosan and its nanocomposite films are shown in Figure 5c. The DSC analysis is a measure of the miscibility and compatibility of the components present in the polymer. The chitosan film showed two major peaks. One was observed in the range of 25 to 150 °C with a maximum at ~88 °C, and this endothermic peak of chitosan was associated with the evaporation of bound and absorbed water [60]. Another peak is shown in the range of 255–320 °C with a maximum of ~286 °C, which might be associated with the thermal degradation of amine units of chitosan. In the case of the chitosan-based nanocomposite films, alteration in both peaks was witnessed. The minor variations of the first peak’s position in the nanocomposite films were conceivably due to the different polymer–water interactions. The difference in the TA-cross-linked and TiO_2_-added films’ peak positions was due to the variable polymer filler interactions. The observed results suggest decent biocompatibility between chitosan and TA/TiO_2_.

#### 3.2.5. Water Vapor Barrier Properties

The WVP of neat chitosan and chitosan-based nanocomposite films is shown in Table 1. The WVP of the neat chitosan film was 0.59 ± 0.03 × 10^−9^ g·m/m^2^·Pa·s, which decreased to 0.51 ± 0.03 × 10^−9^ g·m/m^2^·Pa·s after adding TA, and the WVP significantly further decreased to 0.46 ± 0.03 × 10^−9^ g·m/m^2^·Pa·s after incorporation of 1 wt% of TiO_2_. The resulting insight indicates that the cross-linker (TA), as well as reinforcement (TiO_2_), effectively altered the WVP of the chitosan-based nanocomposite films. The reduction of WVP after adding TA has been reported [26]. Previously, it was also shown that the addition of TiO_2_ decreased the WVP of biopolymer-based nanocomposite films [43]. In addition, in the case of the TiO_2_-added chitosan-based film, a similar improvement in water vapor barrier properties was reported previously [43]. The reduction in WVP (increase in the vapor barrier property) was probably due to the creation of a tortuous path for vapor diffusion through the film by decreasing the free -OH groups [61].

#### 3.2.6. Hydrodynamic and Mechanical Properties

The MC, WS, and SR of neat chitosan and its nanocomposite films are presented in Figure 6a–c. The chitosan-based films’ MC was significantly increased (~12% to ~18%) in all nanocomposite films compared to the pristine chitosan film. In general, the cross-linking reduces the water interaction with the matrix polymer and affects the water retention inside the films. In contrast to the present observation, it was previously reported that cross-linking can effectively reduce the moisture sensitivity of biopolymer-based films [62]. The probable cause of the noteworthy rise in the MC of the films is due to the reduced polymer network interaction as a result of the availability of increased free -OH groups and, consequently, the greater amount of water absorbed [63]. Like the current outcome, previously, MC was increased after addition of nanofillers in chitosan-based films [15]. The SR of the neat chitosan film was drastically decreased (~300% to ~100%) in the case of all the nanocomposite films. The reduction in SR of the nanocomposite film is perhaps due to the blending of hydrophobic nanofiller in the chitosan [63]. The decrease in the chitosan-based film’s swelling degree upon cross-linking was previously reported [25], which validates the present observation. In the biopolymer-based film, the SR mainly depends on the cross-linking material’s porosity and nature [15]. Upon increasing the cross-linking, the SR generally decreases [64], and consequently, the incorporation of TA decreases the SR of chitosan-based nanocomposite films. The WS of the chitosan-based nanocomposite film decreased after adding TA, whereas the addition of TiO_2_ did not much change the WS of the nanocomposite film compared to that of the neat chitosan film. The decrease in WS of the nanocomposite film with TA was possibly due to the cross-linker, TA [26]. In the TA-cross-linked chitosan nanocomposite film, similar WS variation was previously noticed. The WS of the chitosan-based film was found to be affected by the kind and nature of the fillers. The cross-linking reduced the WS, which is rational, whereas the addition of nanofillers did not much influence or slightly increased the WS, which is probably due to the reduction in polymer network interaction in the presence of nanofillers.

The mechanical properties of the chitosan-based nanocomposite films are shown in Figure 7a–d. The average thickness of the chitosan and its nanocomposite films was 20 ± 4 μm. The fabricated films’ thickness was comparable with the thickness (26~30 μm) detected from the cross-section FESEM image and analyzed with ImageJ (Figure 4e–h). The mechanical properties were altered by incorporating TA as a cross-linker and TiO_2_ as a reinforcing nanofiller. The TS, YM, and EB of the neat chitosan film were 95.5 ± 7.7 MPa, 8.8 ± 1.2 GPa, and 2.5 ± 1.2%, respectively, like the previously published data [15]. The TA-cross-linked chitosan-based nanocomposite film’s mechanical properties were significantly improved (TS = 108.8 ± 13.6 MPa) due to TA’s presence. Previously, it was already shown that the cross-linker TA in chitosan improved the composite film’s tensile strength, which agrees with the present findings [26,39]. The mechanical properties were also known to be enhanced by the addition of TiO_2_ in biopolymer-based composite films [43]. In addition, in the case of the TiO_2_-added chitosan-based film, a similar improvement in water vapor barrier properties was reported before [65]. The addition of 0.5 wt% of TiO_2_ improved the TS and YM of the cross-linked chitosan-based film more compared to the addition of 1 wt%. The increase in mechanical properties of the film is predominantly due to the increase in molecular interaction between chitosan and TiO_2_ as well as the good interfacial interactions between them. Previously, it was also observed that the addition of a lower content of nanofiller improved the mechanical performance of the nanocomposite film, whereas at a higher content, it starts decreasing depending on the range [61]. In this work, the synergistic effect of TA/TiO_2_ was considered for better improvement of the mechanical properties of chitosan-based nanocomposite films. As speculated, the addition of TiO_2_ in chitosan-based nanocomposite films significantly improved the mechanical properties, with ~40% improvement in the TS in Chs/TA/Ti^0.5^. In contrast, for Chs/TA/Ti^1.0^, the mechanical properties were reduced, but were still higher than those of the neat chitosan or Chs/TA film. At a 1% TiO_2_ concentration, the chitosan-based film’s mechanical properties were significantly decreased compared to 0.5%, which might be due to the limited interaction between the chitosan and TiO_2_ associated with the phase separation by the aggregated fillers in the polymer matrix. Previously, it was also reported that the addition of TiO_2_ in the biopolymer matrix significantly enhanced the film’s mechanical properties [43]. Zhang et al. [66] also reported that the addition of TiO_2_ in chitosan-based films improved the mechanical properties. The chitosan-based composite film results indicate that incorporating TA/TiO_2_ alone or in combination can significantly improve the chitosan-based nanocomposite films’ mechanical properties, which might be useful for packaging applications.

#### 3.2.7. Antioxidant Properties

Chitosan’s antioxidant actions, as well as those in the nanocomposite films, were determined using the two well-established DPPH and ABTS radical scavenging activity methods, and the obtained results are shown in Figure 8. The neat chitosan film showed considerable antioxidant activity, which is reasonable, as it is an active antioxidant biopolymer. This is due to the hydroxyl and amino groups in chitosan, which can interact with free radicals and show radical scavenging activity [67,68]. The addition of TA in the chitosan-based nanocomposite films showed an enhanced antioxidant action, which might be due to tannic acid’s antioxidant potential [69]. The DPPH and ABTS radical scavenging efficiencies of the neat chitosan film were 17.9% and 30.7%, respectively; meanwhile, they were enhanced to 42.4% and 64.5% in the case of the TA-incorporated chitosan-based film, Chs/TA. The nanocomposite films’ increased antioxidant activity is mainly due to the potent antioxidant tannic acid, which contains a polyphenol group [69]. In the case of TiO_2_-added chitosan-based nanocomposite films, the antioxidant activity was slightly lower than that of the Chs/TA film due to the antioxidant presence of inactive TiO_2_. However, the antioxidant results are similar, indicating that the developed Chs/TA/TiO_2_ nanocomposite films have good antioxidant potential. It has been known that the antioxidant potential of biopolymer-based films is predominantly reliant on the concentration of antioxidants in the films [70].

## 4. Conclusions

Tannic-acid-cross-linked and TiO_2_-reinforced chitosan-based nanocomposite films prepared with the solution casting process are reported. The cross-linked and nanofiller-added chitosan-based films were compatible and were uniformly spread in the polymer matrix. The developed nanocomposite films were see-through and had excellent UV-light barrier properties, without sacrificing too much transparency. The mechanical properties were meaningfully enhanced due to TA’s cross-linking and the reinforcement of TiO_2_. The films’ thermal stability was also slightly improved, whereas the swelling ratio decreased, and the water solubility did not change much. The addition of fillers and a cross-linker slightly improved the water vapor barrier properties of the chitosan-based film. The synergistic effect of TA/TiO_2_ was considered for better improvement of the physical (mechanical, water vapor barrier) properties of the chitosan-based nanocomposite films. Furthermore, the chitosan/TA/TiO_2_ nanocomposite films also showed an intense antioxidant activity. The developed chitosan-based nanocomposite films are promising for active food packaging applications based on the improved mechanical properties, UV barrier properties, and antioxidant behavior.

## Data Availability

Data available on request due to restrictions eg privacy or ethical.

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
