# Peer review of "Tannic-Acid-Cross-Linked and TiO2-Nanoparticle-Reinforced Chitosan-Based Nanocomposite Film"

_polymers, 2021, doi:10.3390/polym13020228_

Round 1

Reviewer 1 Report

This paper reports chitosan-based nanocomposite in film format, with either tannic acid (TA) or the combination of TA and TiO2. The UV, thermal, H2O barrier and mechanical properties of the films are compared, in conjuncion with the antioxidant properties. It is often stated that the largest effect is observed for most properties already with the incorporation of TA, without significant changes for the Chs/TA/Ti systems. Given the very similar thermomechanical properties as presented in figure 5 (the error bars are quite large), I find difficult to justify the purpose of incorporating the TiO2 particles and the advantage gained. Therefore I'm afrain that the work would be of limited interest for the readers of Polymers.

Minor points:
Line 26: Protection (uppercase "P")
Line 171: Check font size of the word "and".
Line 173: Spell out first instance of "FESEM" (field emission...)

Reviewer 2 Report

Dear Editor

In the current article, Tannic acid and TiO2 have used to improve certain properties of chitosan based films. The topic is very important and in the current study it has been handled excellently. The novelty is fine. 

But I have few concerns that needs to be addressed before further processing the article.

-L29, Mucilage is also among the emerging bioplastics and reported in a number of literatures. The authors here have completely missed it. I would suggest to support this line with references below.

-https://www.sciencedirect.com/science/article/pii/S0141813019310918

-https://www.sciencedirect.com/science/article/pii/S0268005X18314358

-L38, please remove "A" from the start of the sentence. Also I would recommend to support the statement by using some examples, such as chitin nano whiskers, cellulose nano fibers, essential oils etc. For convenience I am putting few reference which I would reccomend to discuss here.

-https://www.sciencedirect.com/science/article/pii/S0141813017308917

-https://www.sciencedirect.com/science/article/pii/S0141813017321992

-L85, what do you mean by "using a doctor" please elaborate

-L81, why 0.5% acetic acid was used, why not 1%. please support your statement by a proper reference or explain with a proper reason.

-L83, did the author applied any temperature during sanitation or it was carried out at room temperature, please give a proper explanation for the sake of reproducibility.

-Figure 1, the labeling in figure is very small and poor, please improve the figure. Also, properly combine the figure to present as a whole, in current form it is not good.

-L199-200, the authors are telling that adding TA completely blocked the UV light, which is good. But what about the appearance of the film, as I see from the figures the appearance changes to brownish compared to neat film. What are the author's thought about this, because in packaging the appearance of the material is an important criterion especially food or fruits.

-Figure 4c, please label the direction of endo and exo peaks on y-index.

-Figure 5, please improve the labeling, in current form it is very small and nearly invisible.

-Is it possible to do a cytotoxic analysis for the films or any other assay to see its toxic effects, if any.

-I would also recommend to conduct a water and soil degradability test for the films. For detailed protocol please see the articles which is referred in first comment.

-Please re-write the conclusion section, in current format it is not reflective of the whole study.

Reviewer 3 Report

This paper reports an investigation about the reinforcement of chitosan-based films. Authors evaluated the influence of tannic acid crosslinking, as well as TiO2 nanoparticles through mechanical, thermal and antioxidant characterization. The paper includes interesting results with suitable experimental design, data analysis and discussion. Therefore, it is recommended for publication in Polymers after major revision indicated below.

INTRODUCTION

  • A paragraph about the processing technique selected should be included, considering previous studies.

MATERIALS AND METHODS

  • Please include the deacetylation degree of chitosan.
  • Could you better explain this statement: MW: 190,000-310,000 based on viscosity [line 74]. How was the MW of chitosan measured?
  • In line 112, there is a “w” missing (where instead of here)

RESULTS AND DISCUSSION

  • The symbol used for chitosan (Chs) should be included once chitosan is described in the Materials and Methods Section.
  • The scale bar in Figure 1a cannot be seen properly.
  • The word “saccharide” appears in another format [line 215]. Please correct it.
  • Compare the thickness obtained with the micrometer with the cross-sectional images of the films obtained (Figure 3).
  • Revise the letters included in Figure 5d since I think there is no significant differences between Chs/TA/Ti5 and Chs/TA/Ti1.0 systems.
  • In Section, authors only described the evolution found in Figure, but without further explanation of the reasons of the changes in the hydrodynamic and the mechanical properties of the different films with TA or TiO2

REFERENCES

  • Authors should follow the format for the References Section according to the Instructions for authors.

Round 2

Reviewer 1 Report

The new version is substantially improved, providing adequate purpose & background information, as well as revised interpretation. I think that now it is suitable for publication.

Reviewer 2 Report

The article is ready for publication. All my comments are addressed.

Reviewer 3 Report

Authors performed all the corrections during the review process. Therefore, I recommend this manuscript for publication.